# Comparing Assessment Tools as Candidates for Personalized Nutritional Evaluation of Senior Citizens in a Nursing Home

**DOI:** 10.3390/nu13114160

**Published:** 2021-11-20

**Authors:** Diogo Sousa-Catita, Maria Alexandra Bernardo, Carla Adriana Santos, Maria Leonor Silva, Paulo Mascarenhas, Catarina Godinho, Jorge Fonseca

**Affiliations:** 1Residências Montepio—Serviços de Saúde, SA, 1600-131 Lisboa, Portugal; 2Grupo de Patologia Médica, Nutrição e Exercício Clínico (PaMNEC) do Centro de Investigação Interdisciplinar Egas Moniz (CiiEM), 2829-511 Almada, Portugal; cgcgodinho@gmail.com (C.G.); jorgedafonseca@hotmail.com (J.F.); 3GENE—Artificial Feeding Team, Hospital Garcia de Orta, 2805-267 Almada, Portugal; carla.adriana.santos@hotmail.com; 4Centro de Investigação Interdisciplinar Egas Moniz (CiiEM), 2829-511 Almada, Portugal; abernardo@egasmoniz.edu.pt (M.A.B.); lsilva@egasmoniz.edu.pt (M.L.S.); pmascarenhas@egasmoniz.edu.pt (P.M.)

**Keywords:** geriatric, aging, old age, nutritional screening, mini nutritional assessment short form, calf girth

## Abstract

Nutrition is an important health issue for seniors. In nursing homes, simple, inexpensive, fast, and validated tools to assess nutritional risk/status are indispensable. A multisurvey cross-sectional study with a convenient sample was created, comparing five nutritional screening/assessment tools and the time required for each, in order to identify the most useful instrument for a nursing home setting. Nutrition risk/status was evaluated using the following tools: Subjective Global Assessment (SGA), Mini Nutritional Assessment Short Form (MNA-SF), Malnutrition Universal Screening Tool (MUST), Nutritional Risk Screening 2002 (NRS 2002), and calf girth (CG). The time spent completing each tool was recorded. Eighty-three subjects were included. MNA-SF and CG were the screening tools that ranked highest with regards to malnutrition identification. CG failed to identify nutritional risk/malnutrition in seniors with lower limb edema. CG was the fastest tool while SGA was the slowest. This was the first study comparing non-invasive nutritional tools with time expended as a consideration in the implementation. CG is responsive, fast, and reliable in elders without edema. MNA-SF was more efficient at detecting malnutrition cases in the elderly population. Both MNA-SF and CG are considered the most suitable for the nursing home setting.

## 1. Introduction

According to the World Health Organization (WHO), the prevalence of malnutrition risk increases with age [1]. In this context, clinical tools have been developed and validated to evaluate nutritional risk and status [2,3,4,5,6]. Multiple tools include clinical, biochemical, and anthropometric parameters. Some of them are especially focused on the elderly [2] while others were created to be used on the general population [3,4,5,6]. Nursing homes must deal with large numbers of senior citizens, often with a wide range of pathologic conditions and clinical status, from active aging seniors to very dependent older patients. Studies have reported that malnutrition is directly associated with the functional impairment of senior populations, and nutritional status is indirectly associated with the presence and severity of comorbidities [7,8]. Given the diversity and number of senior citizens that require assessment, nursing homes have a need to identify simple, inexpensive, and fast tools to assess nutritional risk and/or status in order to save time for health professional and reduce inconvenience to the senior population [9]. A previous study compared six nutrition assessment tools [10] in order to identify the optimal instrument. However, some of the tools in that study included biochemical parameters that are not practical in nursing home environments due to the costs involved, the need for blood collection, and the extended time needed for the laboratory results before clinical decision-making can occur. For personalized care of senior citizens in nursing homes, it is important to identify widely available tools that do not need much time or require invasive or costly techniques, such as blood samples.

In our experience working with senior citizens, laboratory tests and complex nutritional assessment tools are impractical and should be avoided in nursing home settings. Seniors will be more amenable to nutritional assessment when collaboration is easy, fast, and without invasive maneuvers. To the best of our knowledge, the search for the balance between simplicity and sensitivity of nutritional tools has not been previously addressed in the clinical nutrition literature.

To achieve effective and personalized nutritional evaluation of senior citizens in nursing homes [11], we sought to identify nutritional evaluation instruments that consider age, mental, and physical conditions. The present study evaluates five widespread non-invasive nutrition screening/assessment tools (Mini Nutritional Assessment Short Form (MNA-SF) [2], Malnutrition Universal Screening Tool (MUST) [5], Nutritional Risk Screening (NRS 2002) [3], calf girth (CG) [4], and Subjective Global Assessment (SGA) [6]). Our study aimed to compare a range of options in order to identify the most suitable nutritional tool for the nursing home senior population. We specifically aimed to:Compare the capacity to identify nutritional risk/malnutrition in the nursing home elders.Evaluate the time spent for each tool.

## 2. Methods

### 2.1. Design

We used a multisurvey cross-sectional study. All elder citizens were evaluated in a single observation by the same researcher. The evaluation of all subjects was performed over the course of one month.

### 2.2. Sampling and Recruitment

For the present study, the inclusion criteria were men or women, 65 years old and above, living in or attending a nursing home, and who signed the informed consent. The exclusion criteria included a mini-mental score (MMS) <20, as registered by the institution psychologist, or the presence of psychiatric illnesses, or other factors that could hamper the ability to obtain the anthropometric measurements required [12].

Elderly citizen users in nursing homes located in the periphery area of Lisbon were invited to participate in this study. After exercising the aforementioned exclusion criteria, we obtained a final sample of 83 seniors, including 59 institutionalized subjects and 24 attending adult daycare programming.

The evaluation of each participant was carried out by the same dietitian in a single visit and the instruments were used sequentially: MNA-SF, MUST, NRS2002, SGA, and CG. All participants were evaluated in the same month in the nursing home.

### 2.3. Ethics and Procedures

The study was approved by the Ethics Committee of Egas Moniz, CRL (ID: 7/2015 on 24 March 2015) and was carried out in accordance with the Helsinki Declaration. A signed informed consent was obtained from each participant after written and oral information about the purpose of the study and procedures involved was provided. All data collection was registered through a coded number attributed to each participant.

### 2.4. Anthropometric Parameters

All anthropometric measurements were performed in the morning and the subjects were wearing lightweight clothing suitable for assessment and were without shoes.

A SECA scale, with a range of 4 kg to 360 kg (0.1 kg) and an ADE stadiometer, with a range of 85 to 210 cm (±0.1 cm) were used capture weight and height. Body Mass Index (BMI) was calculated as weight (kg) divided by height (m) squared (kg/m^2^). Seniors with reduced mobility were weighed in their wheelchairs, with their height being ascertained from their Citizen Card. Unintentional weight loss was estimated according to reports from the participants. Anthropometric measurements were assessed three consecutive times by the same investigator and reported values corresponded to mean values.

### 2.5. Nutritional Screening and Assessment

Participants were subjected to all nutritional risk and status screening tools (MNA-SF, MUST, SGA, NRS2002, and CG).

#### 2.5.1. Mini Nutritional Assessment Short Form (MNA-SF)

MNA-SF is a screening tool that assesses six different areas of concern. The Short Form, available on Nestlé’s website, was developed and validated specifically for an older population (>60 years) [2]. Completion of the MNA-SF results in a score with 3 categories: normal (12–14), risk of malnutrition (8–11), and malnourished (0–7).

#### 2.5.2. Malnutrition Universal Screening Tool (MUST)

The MUST is a multistep nutritional screening tool using a model that was developed by the Malnutrition Advisory Group, an integrated committee of the British Association for Parenteral and Enteral Nutrition (MAG-BAPEN) [5]. Completion of the MUST results in a score within 3 categories: low risk (0), medium risk (1), and high risk (>2) [6].

#### 2.5.3. Nutritional Risk Screening 2002 (NRS 2002)

The NRS 2002 is a two-step nutritional screening tool. The NRS 2002 was developed and certified by the European Society for Clinical Nutrition and Metabolism (ESPEN) [3]. Completion of the NRS 2002 results in a score within 2 categories—normal (<3), and nutritional risk (≥3).

#### 2.5.4. Calf Girth (CG)

The outcome from the measurement of calf girth is a score graded in 2 categories—normal (≥31 cm), and nutritional risk (<31) [4]. This technique was used in accordance with the procedures described in the “*Kinanthropometry and Exercise Physiology Laboratory Manual*” [13].

#### 2.5.5. Subjective Global Assessment (SGA)

SGA is a nutritional status assessment tool that incorporates inquiries from a clinician in conjunction with a physical examination. With the SGA we obtain a score with 3 categories: well nourished (A), moderate malnutrition (B), and severe malnutrition (C). The SGA tool was developed and validated by Detsky et al. (1987) [6]. It was performed by a single trained researcher throughout the study.

For the purposes of the study, distinguishing between the normal nutritional status and malnutrition risk (which was categorized as any identification of risk), which resulted in 2 broad categories: normal nutritional status and nutritional risk/malnutrition (Table 1).

### 2.6. Statistical Analysis

Statistical analysis was performed with SPSS^®^ Statistics (Statistical Package for Social Sciences) software version 25.0 for Windows^®^. Data are presented as mean ± SDs or SEMs. All statistical tests were performed at the 5% level of significance.

To evaluate the association and concordance between the various instruments as 2-level ordinal variables (Table 1), we computed Kendall’s tau-b correlation coefficient and Kendall’s W coefficient of concordance, respectively.

Kendall’s tau-b was used as a pairwise association measure, ranging between (−1) and (1) with values close to 0 meaning independence of the variables. On the other hand, the coefficient of concordance, Kendall’s W measured the overall agreement for nutritional risk/malnutrition among the tools. Kendall’s W ranges between 0 (no agreement) and 1 (complete agreement) and, for each tool, mean ranks for malnutrition assessment are computed. Finally, logistic ordinal generalized linear models were fit to evaluate the effect of age, sex, MMS and group (institutionalized/day care) on the screening tool outcomes.

## 3. Results

A total of 120 subjects, including institutionalized and day care populations were invited to participate in the study. After verification of inclusion and exclusion criteria, 83 subjects were included in the study: men 38.6% (*n* = 32) and 61.4% (*n* = 51) women (Figure 1). Most subjects were Caucasian, with only five of African ethnicity (three males and two females).

Participant’s ages ranged between 65 to 100 years old, and the mean was 78.5 ± 8.7 years. The mean height was 156 ± 9 cm, the mean weight was 64.5 ± 12.5 kg and the mean BMI was 26.7 ± 5.2 kg/m^2^ (range from 17 kg/m^2^ to 44 kg/m^2^).

### Characterization of Nutritional Status

Four subjects had lower limb edema that can misinform measurements in CG. Nevertheless, these four measurements were ultimately included in our evaluation.

We divided the nursing home population in two groups: 1—institutionalized and 2—day care. Institutionalized participants presented with higher prevalence of nutritional risk/malnutrition than the day care participants in all tools, with the exception of the MUST. We highlight these results in Figure 2 and Figure 3.

For the global population of the nursing home, MNA-SF obtained the higher prevalence of nutritional risk/malnutrition (34.9%). The lower percentage of nutritional risk/malnutrition was obtained with NRS 2002 (8.4%) (Figure 4).

Kendall’s tau-b correlation coefficient showed a positive and statistically significant correlation with all the instruments (Table 2).

The time spent on applying the CG was the shortest (40 s), followed by MNA-SF (2 min), NRS 2002 and MUST (3 min) and the SGA was the longest (9 min) (Table 3).

The Kendall’s W value of 0.15 reflected a poor agreement between the assessment tools, and MNA-SF was the one that performed most favorably (mean rank = 3.35) for nutritional risk/malnutrition identification (Table 4). Age, sex, and MMS did not show significant adjusted effects on the tool’s outcome, however, there were significant differences in nutritional risk/malnutrition incidence between institutionalized and day care groups when we applied MNA-SF and CG adjusted for age, sex and MMS (Table 5). Therefore, these results suggest that, among the applied tools, MNA-SF and CG have increased power to discriminate nutritional risk/malnutrition rate differences between two groups of subjects.

## 4. Discussion

In this study we confirmed that the overall prevalence of malnutrition in senior’s nursing homes is high, as detected by all the nutritional screening/assessment tools used. This result is similar with other studies involving nursing homes for institutionalized senior citizens showing that nutritional personalized care is needed [14,15] and should be provided according to the results of screening/assessment tools, comorbidities, preferences, and habits of senior citizens. As expected, institutionalized elders present with a higher prevalence of nutritional risk/malnutrition compared to the day care population even when all tools are adjusted to age, sex, and MMS. This result occurs in most nursing homes within the same clinical demographic [16,17]. This evidence can be explained by the types of issues previously described, including that institutionalized elders frequently have a chronic condition with more disabilities/co-morbidities, experience more social problems, and have a less active lifestyle when compared to day care elders [18]. Based on the current research, those aspects may contribute to diminish appetites, alimentary problems, and weight loss and consequently to malnutrition [11,19].

The prevalence of nutritional risk/malnutrition was higher in the institutionalized than in the day care population in all tools, except for MUST. This can be explained, as this tool was created as a broad-spectrum tool for communities and hospitals [5]. It may have a better capacity to identify malnutrition in day care seniors but not in the fully institutionalized.

To the best of our knowledge, our study is the first to compare non-invasive nutritional screening/assessment tools with a consideration of the time necessary to complete as a metric of their practicability. There is currently no gold standard that we could rely on to define sensitivity and specificity. Nonetheless, tools that are sensitive enough to identify larger numbers of malnourished elder citizens in a time-efficient manner are of the utmost importance. We believe that in the clinical context of senior citizens, it is preferable to possibly over diagnose malnutrition than fail to identify possible instances of malnutrition. Notably, most of the tools are screening tools for assessment of nutritional risk rather than the actual presence of malnutrition. Only the SGA is a complete nutritional assessment tool [20]. The main disadvantages of the SGA are the long time it requires to be completed and the extensive experience required from the professional to apply it. As a result, the most appropriate nutritional screening/assessment tool in this study sample was MNA-SF, which agrees with previous studies on this topic [2]. Moreover, MNA-SF only required a short amount of time to administer. Notably, the measurement of BMI was a parameter that represented the most difficulty in its application, especially when taking into account the bedridden nature of patients included for evaluation in seniors nursing homes. Furthermore, the neuropsychological evaluation needed for screening, as in item E (classified as normal, mild dementia, or severe dementia) could also be difficult to evaluate for health professionals with no specific training in the neuropsychological field. The MUST assessment tool is a quick and easy application tool and, according to BAPEN, this tool can be applied to the elderly population with success [5]. However, the results of the present study are not in agreement since the MUST tool did not prove to be an effective and efficient tool to identify nutritional risk/malnutrition in our clinical setting.

The NRS 2002 assessment tool was also quick and easy to apply but was the least effective and efficient tool. Despite being certified by ESPEN [3] and validated for adults, including additional validation for individuals aged over 70 years, the NRS 2002 tool was developed for hospitalized patients. In this context, it is plausible to consider that it may not be so accurate for identifying malnutrition in a non-hospital setting.

In our study, the CG assessment tool proved to be the quickest and easiest application tool, being a very effective and efficient tool of nutritional risk/malnutrition identification. The CG tool’s major handicap was difficulty identifying nutritional risk/malnutrition in subjects with lower limb edema. Edema influences the measurement values, classifying nutritional status as normal in elders that may present nutritional risk/malnutrition. This tool cannot be recommended for seniors with lower limb edema. Conversely it may be very useful in seniors with reduced cognitive ability, but since these users were excluded by our study criteria, this advantage could not be demonstrated, and therefore remains a suggestion only pending the results of future studies. The results of the present study demonstrate that MUST, NRS 2002, and SGA presented low capacity for identifying malnutrition, and we believe that they are not the best choices for use in nursing home settings.

Globally, CG is easy to evaluate proving to be very fast, competent, and reliable if individuals have no lower limb edema. MNA-SF is a very competent tool to detect nutritional risk/malnutrition and is fast to apply, providing the individuals have preserved cognitive skills.

When all tools were adjusted to several different variables like age, sex, and MMS score, the tool results remained similar, suggesting that these variables did not affect the nutritional evaluation results.

Our study presents some limitations. It was a single center study, with a limited sample size and did not consider any tool with laboratory parameters, as in most nursing homes there is no option to do blood sampling at regular intervals because it is invasive for the elderly and expensive. Nevertheless, laboratory data included in other tools may provide interesting information that we did not assess. We aggregated each tool into only two categories: “nutritional risk/malnutrition” and “normal nutritional status”, as used in earlier studies published in referential nutrition journals [10,21], but it is dependent on the tools used, and some discrimination capacity may have been lost.

## 5. Conclusions

Considering the high prevalence of malnutrition in seniors nursing homes, it is important to identify a simple, fast, validated, cheap, and non-invasive nutritional screening tool for routine evaluation. This is imperative in order to improve nutritional care and to guarantee personalized nutritional interventions. Drawing from the present study, we advocate the use of MNA-SF in nursing home seniors with preserved cognitive skills. The CG seems to be the more adequate choice for seniors with impaired cognitive skills and no limb edema.

## Figures and Tables

**Figure 1 nutrients-13-04160-f001:**
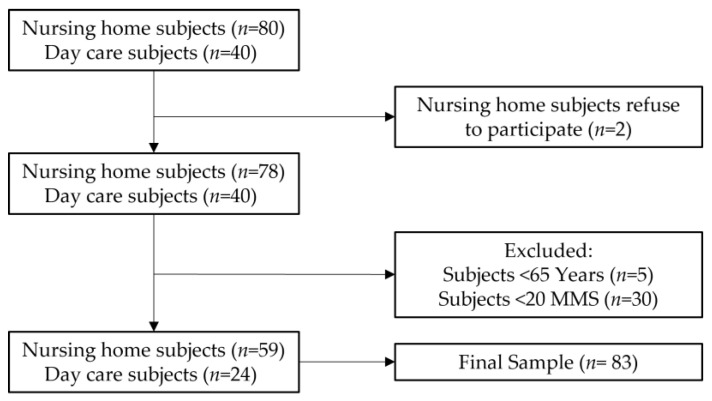
Sample flowchart of the cross-sectional study.

**Figure 2 nutrients-13-04160-f002:**
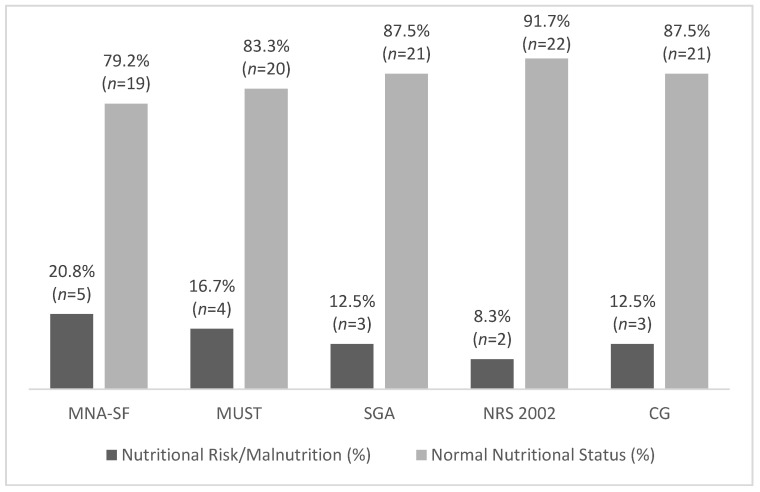
Percentage mean value obtained from each nutritional assessment and screening tool in day care subjects (MNA-SF—Mini Nutritional Assessment Short Form; MUST—Malnutrition Universal Screening Tool; SGA—Subjective Global Assessment; NRS 2002—Nutritional Risk Screening 2002; CG—calf girth).

**Figure 3 nutrients-13-04160-f003:**
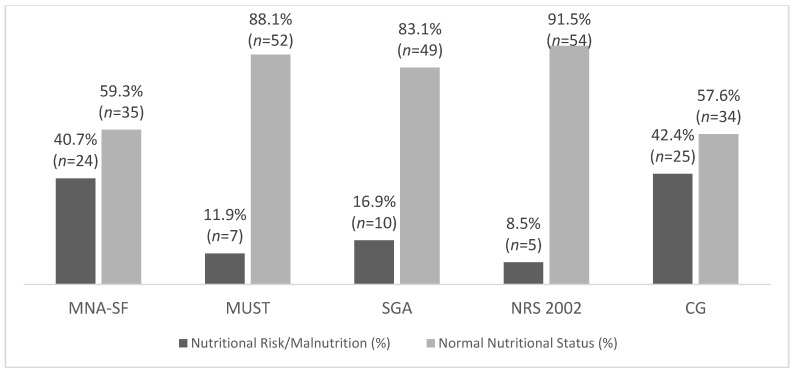
Percentage mean value obtained from each nutritional assessment and screening tool in the institutionalize population (MNA-SF—Mini Nutritional Assessment Short Form; MUST—Malnutrition Universal Screening Tool; SGA—Subjective Global Assessment; NRS 2002—Nutritional Risk Screening 2002; CG—calf girth).

**Figure 4 nutrients-13-04160-f004:**
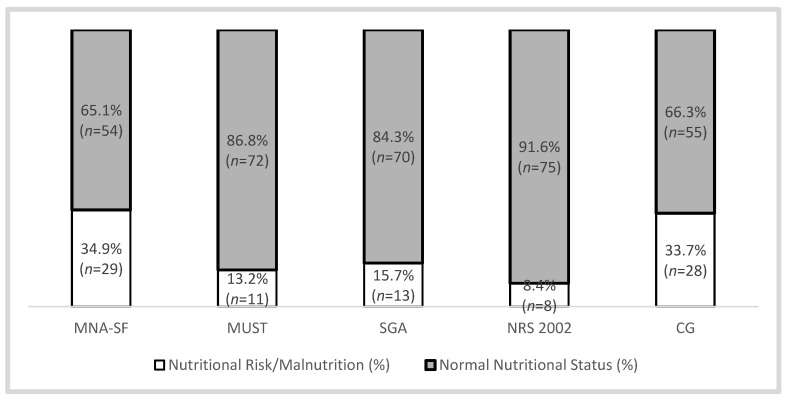
Percentage mean value obtained from each nutritional assessment and screening tool in nursing home elders (MNA-SF—Mini Nutritional Assessment—Short Form; MUST—Malnutrition Universal Screening Tool; SGA—Subjective Global Assessment; NRS 2002—Nutritional Risk Screening 2002; CG—calf girth).

**Table 1 nutrients-13-04160-t001:** Sample nutritional status classified in two categories, even if the tool implemented typically incorporated 3 categories.

	Nutritional Assessment Tool				
	MNA-SF	MUST	SGA	NRS 2002	CG
Normal Nutritional Status	12 to 14	0	A	0 to 2	≥31
Nutritional Risk/Malnutrition	0 to 11	≥1	B and C	≥3	<31

MNA-SF—Mini Nutritional Assessment Short Form: score 12–14: normal, 8–11: risk of malnutrition, and 0–7: malnourished; MUST—Malnutrition Universal Screening Tool: score low risk (0), medium risk (1), and high risk (>2); SGA—Subjective Global Assessment: score well nourished (A), moderate malnutrition (B), and severe malnutrition (C), NRS—Nutritional Risk Screening 2002: score normal (<3) and nutritional risk (≥3); CG—calf girth: score normal (≥31 cm) and nutritional risk (<31).

**Table 2 nutrients-13-04160-t002:** Statistical comparison of nutritional assessments and screening tools.

	MNA-SF	MUST	SGA	NRS2002
MUST	0.38 *			
SGA	0.45 *	0.71 *		
NRS2002	0.32 *	0.65 *	0.59 *	
CG	0.49 *	0.40 *	0.46 *	0.24 **

MNA-SF—Mini Nutritional Assessment Short Form; MUST—Malnutrition Universal Screening Tool; SGA—Subjective Global Assessment; NRS 2002—Nutritional Risk Screening 2002; CG—calf girth. * *p* < 0.001, ** *p* < 0.05.

**Table 3 nutrients-13-04160-t003:** Average time of each nutritional tool assessment spent with one subject.

Average Time (± SD)	**MNA-SF**	**MUST**	**SGA**	**NRS 2002**	**CG**
	2 min (±0.52)	3 min (±0.85)	9 min (±1.14)	3 min (±0.57)	40 s (±2.64)

MNA-SF—Mini Nutritional Assessment—Short Form; MUST—Malnutrition Universal Screening Tool; SGA—Subjective Global Assessment; NRS 2002—Nutritional Risk Screening 2002; CG—calf girth.

**Table 4 nutrients-13-04160-t004:** Mean rank of each nutritional tool assessment for nutritional risk/malnutrition identification.

Kendall’s Mean Rank	**MNA-SF**	**MUST**	**SGA**	**NRS 2002**	**CG**
	3.35	2.81	2.87	2.66	3.32
Kendall’s W	0.15 *				

MNA-SF—Mini Nutritional Assessment—Short Form; MUST—Malnutrition Universal Screening Tool; SGA—Subjective Global Assessment; NRS 2002—Nutritional Risk Screening 2002; CG—calf girth. * *p* < 0.001.

**Table 5 nutrients-13-04160-t005:** Odds ratios (OR) for nutritional risk/malnutrition identification.

OR (*p*-value) Age	**MNA-SF**	**MUST**	**SGA**	**NRS 2002**	**CG**
	0.999 (0.980)	1.015 (0.729)	1.032 (0.445)	1.048 (0.427)	0.983 (0.551)
Sex (Female)	0.325 (0.172)	1.417 (0.648)	1.882 (0.391)	2.556 (0.417)	1.035 (0.950)
MMS	0.829 (0.232)	0.811 (0.359)	0.811 (0.315)	0.952 (0.866)	0.814 (0.175)
Group (Day Center)	0.325 (0.049) *	1.216 (0.794)	0.563 (0.463)	1.253 (0.824)	0.146 (0.008) *

MNA-SF—Mini Nutritional Assessment Short Form; MUST—Malnutrition Universal Screening Tool; SGA—Subjective Global Assessment; NRS 2002—Nutritional Risk Screening 2002; CG—calf girth; MMS—mini-mental score. * *p* < 0.05.

## Data Availability

The data presented in this study are available on request from the first author.

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
