# Peer review of "Comparing Assessment Tools as Candidates for Personalized Nutritional Evaluation of Senior Citizens in a Nursing Home"

_nutrients, 2021, doi:10.3390/nu13114160_

Round 1
Reviewer 1 Report
dear authors, thank you for this paper.
Screening is a process for evaluating the possible presence of a particular problem. The outcome is normally a simple yes or no.
Assessment is a process for defining the nature of that problem, determining the magnitude of the problem, and developing specific solution recommendations for addressing the problem.
thus title needs to mention only assessment (pls delete screening).
keyword: aging, old age, elderly will increase the visibility of search avoid repeating words from title
introduction give a brief description of the scientific background and rationale behind the reported investigation on each elements this introduction is very scarce and not clear
State specific objectives, including any prespecified hypotheses
methods
Include the dates of recruitment, exposure, follow-up, and data collection, along with the setting, locations, and dates of recruitment, exposure, and follow-up
who did the mini ms (MMS)? who performed all testing. please give details.
Give the eligibility criteria, and the sources and methods of
selection of participants
Clearly define all outcomes, exposures, predictors, potential confounders, and effect modifiers. Give diagnostic criteria, if applicable
data analysis plan is very primitive and basic
Give characteristics of study participants (eg demographic, clinical, social) and information on exposures and potential confounders
Describe any sensitivity analyses
Report other analyses done—eg analyses of subgroups and interactions and sensitivity analyses
speak with biostatistician to develop regression analyses to control for age sex and MMS
results need to be developed on basis of the
results need to be improved based on newer analyses
discussion and conclusion to be improved based on result of the newer analyses
Reviewer 2 Report
Very interesting and needed study. Major issue with the manuscript is that there is no control/gold standard tool to compare the tools under investigation.
Other minor comments are in the attached document.

Round 2
Reviewer 1 Report
dear authors, thanks for addressing my concerns.
Author Response
We are the ones who appreciate all the suggestions for improvement . Thank you very much